# SaGe-Path: Pay-as-you-go SPARQL Property Path Queries Processing using Web Preemption

Julien Aimonier-Davat,[1] Hala Skaf-Molli[1], and Pascal Molli[1]

LS2N, University of Nantes, Nantes, France
{julien.aimonier-davat,hala.skaf,pascal.molli}@univ-nantes.fr

**Abstract.** SPARQL property path queries allow to write sophisticated navigational queries on knowledge graphs (KGs). However, the evaluation of these queries on online KGs are often interrupted by fair use policies, returning only partial results. SaGe-Path addresses this issue by relying on the concept of Partial Transitive Closure (PTC). Under PTC, the graph exploration for SPARQL property path queries is limited to a predefined depth. When the depth limit is reached, frontier nodes are returned to the client. A PTC-client is then able to reuse frontier nodes to continue the exploration of the graph. In this way, SaGe-Path follows a pay-as-you-go approach to evaluate SPARQL property path queries. This demonstration shows how queries that do not complete on the public Wikidata SPARQL endpoint can complete using SaGe-Path. An extended user-interface provides real-time visualization of all SaGe-Path internals, allowing to understand the approach overheads and the effects of different parameters on performance. SaGe-Path demonstrates how complex SPARQL property path queries can be efficiently evaluated online with guaranteed complete results.

**Keywords:** Semantic Web, SPARQL property path queries, web preemption

## 1 Introduction

SPARQL property path queries provide a succinct way to write complex navigational queries over RDF knowledge graphs. However, the evaluation of these queries on online knowledge graphs such as DBPedia or Wikidata is often interrupted by quotas, returning no results or partial results. For example, the query $Q_1$ in Figure 1 which returns creative works with the list of fiction works that inspired them is killed after 60s by the public Wikidata SPARQL endpoint. Consequently, only partial results are returned.

```
PREFIX wd: <http://www.wikidata.org/entity/>
PREFIX wdt: <http://www.wikidata.org/prop/direct/>
SELECT ?creativeWork ?fictionalWork  WHERE {
   ?creativeWork  wdt:P144  ?fictionalWork  .
   ?creativeWork  wdt:P31/wdt:P279* wd:Q17537576  .
   ?fictionalWork  wdt:P136  wd:Q8253}
```

Fig. 1: $Q_1$: Creative works and the list of fictional works that inspired them on Wikidata

It is possible to get rid of quotas by evaluating path queries with Triple Pattern Fragment (TPF) [5, 3] or Web Preemption [1]. However, as these approaches have no support for transitive closures on server-side, they materialize transitive closures on client-side. Then they perform joins with the other part of the query, generating a huge data shipping that drastically degrades performances. For instance, SaGe is able to terminate query $Q1$ but requires more than 17000 HTTP calls to complete with an execution time of  1100s.

In [2], we presented a new approach to evaluate SPARQL property path queries by introducing the concept of Partial Transitive Closure (PTC). The PTC of a property path query $Q$ corresponds to the evaluation of $Q$ with a limited exploration depth. We demonstrated that a preemptable

server [4] is able to fairly execute the PTC of any Basic Graph Patterns (BGPs) containing path expressions while returning complete results. When the PTC evaluation reaches the depth limit, the PTC evaluation returns frontier nodes indicating where the graph exploration has been stopped. Thanks to this information, a PTC-client generates new path queries that restart the exploration from these frontier nodes. By iterating over frontier nodes, the client is able to go deeper and deeper in the graph exploration following a pay-as-you-go approach. Compared to the state of art, all transitive closures and BGPs are now executed on the preemptable server, only results and control information are shipped to the clients. As described in [2], SAGE-PATH outperforms previous approaches both in terms of HTTP calls, data transfer and query execution time. For example, SAGE-PATH is able to complete the query $Q_1$ in only 80s and 1500 HTTP calls for a depth set to 5.

For this demonstration, we build an interface that allows users to understand SAGE-PATH internals, how the evaluation of the SPARQL property path queries works, what are the underlying costs, what happens when changing the quanta of the web preemption and the depth limit of the Partial Transitive Closure. Finally, we show how SPARQL property path queries that cannot complete on public SPARQL endpoints can be easily terminated thanks to SAGE-PATH.

## 2 Overview of SAGE-PATH

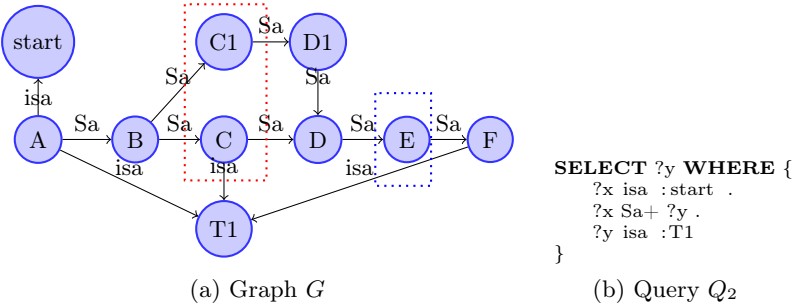

(a) Graph $G$

```
SELECT ?y WHERE {
    ?x isa  : start  .
    ?x Sa+ ?y .
    ?y isa  : T1
}
```

(b) Query $Q_2$

Fig. 2: Graph $G$ and Query $Q_2$

SaGe-Path extends SaGe [4] with a PCT Operator on the server-side. The PTC of a property path query $Q$ corresponds to the evaluation of $Q$ where the exploration of the graph for transitive closures is limited to a maximum depth $d$. Under PTC, the evaluation of a property path query returns not only solution mappings, but also visited nodes with their corresponding depths. Nodes visited at a depth equal to $d$ are called *frontier nodes*.

To illustrate, consider the simple graph $G$ defined in Figure 2a and the query $Q_2$ in Figure 2b. Under a PTC with $d = 2$, the evaluation of $Q_2$ over $G$ returns the solution $\{?y \mapsto C\}$ and the set of visited nodes $\{(B, 1), (C, 2), (C1, 2)\}$ with their depths. Because $C$ and $C1$ have been reached at depth 2 (red dotted rectangle in Figure 2a), they are both marked as frontier nodes.

Because $Q_2$ is executed under web preemption, it may require several quanta to terminate. In this case, it is possible to visit the same nodes several times, more details in [2]. We refer to these nodes as duplicates, this is clearly an overhead of the SAGE-PATH approach. Small quanta are more likely to generate duplicates.

Query $Q_2$ terminates with 2 frontier nodes $C$ and $C1$. To continue the exploration of the graph $G$, two new queries $Q21$ and $Q22$ are generated as described in Figure 3. The execution of $Q21$ returns no results but two new visited nodes $\{(D1, 1), (D, 2)\}$ where $D$ is a frontier node. The evaluation of $Q22$ also returns no results but new visited nodes $\{(D, 1), (E, 2)\}$ where $E$ is a frontier node. As we can see, $D$ has been visited twice, leading to the transfer of duplicates. However, because $Q22$ reached $D$ at a lowest depth than $Q21$, after the execution of $Q22$, $D$ is no longer considered

| Q21: **SELECT** ?y **WHERE** { | Q22: **SELECT** ?y **WHERE** { | Q221: **SELECT** ?y **WHERE** { |
|---|---|---|
| BIND(:**A** as ?x).
:C1 Sa+ ?y.
?y isa :T1
} | BIND(:**A** as ?x).
:C Sa+ ?y.
?y isa :T1
} | BIND(:**A** as ?x).
:E Sa+ ?y.
?y isa :T1
} |

Fig. 3: Expanded path queries

as a frontier node and will not be expand by the client. Consequently, only $E$ remains a frontier node (blue dotted rectangle in Figure 2a). A new query $Q221$ is generated from $Q22$ to expand the node $E$ (cf Figure 3). The execution of $Q221$ returns a new solution mapping $\{?y \mapsto F\}$ and a new visited node $(F, 1)$. As all frontier nodes have been explored, the query $Q_2$ terminates. By merging results of $Q2$, $Q21$, $Q22$ and $Q221$, i.e. $\{?y \mapsto C\}$ and $\{?y \mapsto F\}$, we obtain complete results.

As we can see, PTC follows a pay-as-you-go approach that returns results with the shortest path first modulo $d$. This can be very interesting in interactive use-cases. Another advantage is that all joins are performed on the server. This is why SaGe-Path outperforms the state of art approaches [1]. These advantages come at the price of transferring visited nodes, perhaps multiple times. This overhead is highly dependent of the exploration depth $d$ and the value of the quantum. The demonstration allows users to understand how these two parameters influence the overhead and the global performance.

## 3   Demonstration scenario

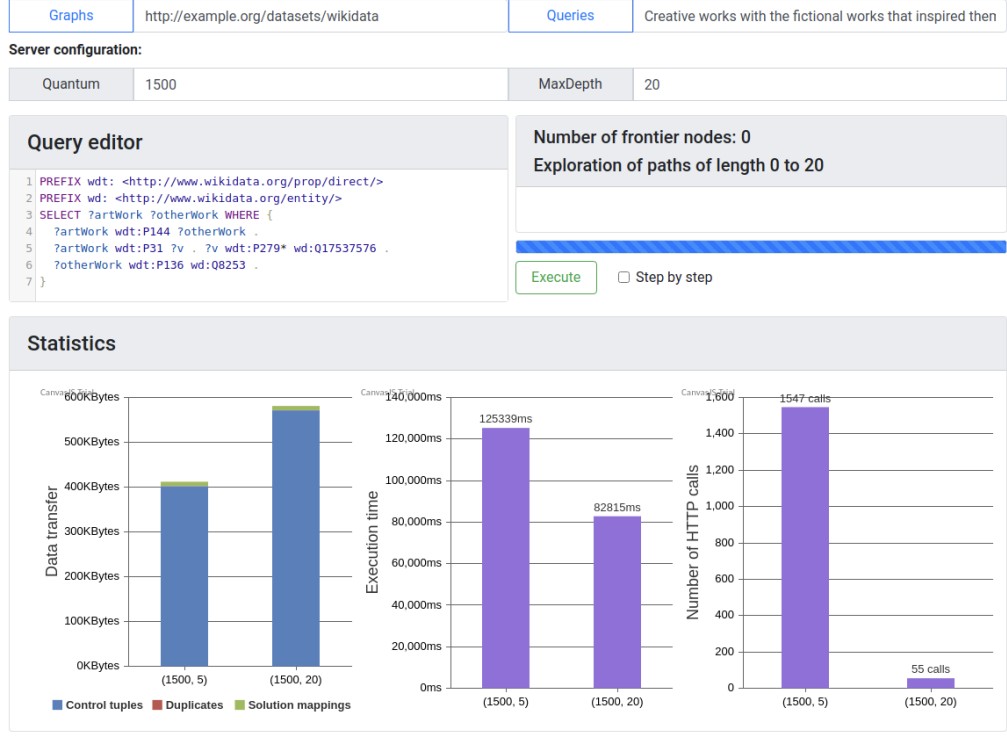

Fig. 4: User-Interface of SaGe-Path demonstration

The scenario is based on queries that do not terminate on the SPARQL endpoint of Wikidata, such as the query $Q1$ (of course, a visiter can try her own property path queries). The demonstration

shows how SaGe-Path is able to complete such queries thanks to the PTC approach. The source-code of the demonstration with a video are available online at [1] and [2].

Figure 4 presents a fragment of the final result of the execution of query $Q1$ with the SaGe-Path user interface. User can change the quantum and the MaxDepth parameters to see the effects on the evaluation of the query. Step-by-step execution allows to see all frontier nodes and expanded queries. Statistics allow to compare different runs for different values of quantum and MaxDepth. In Figure 4, $(1500, 5)$ corresponds to a run of the query with the quantum set to 1500ms and a MaxDepth set to 5. As we can see, running $Q_1$ with a MaxDepth of 5 generates less data transfer than a run with MaxDepth of 20, but the execution time and the number of HTTP calls are significantly increased.

## 4   Conclusion

In this demonstration, we presented SaGe-Path: a pay-as-you-go approach for processing online SPARQL property path queries and get complete results. The visualization of the progress of the query execution allows to observe how SaGe-Path servers provide a partial evaluation of a property path and how SaGe-Path clients use control information sent by the server to continue the execution until complete query execution. Compared to SPARQL endpoints, SaGe-Path always returns complete results and preserves the responsiveness of the server thanks to the web preemption. Compared to client-side approaches, SaGe-Path drastically improves performance in term of execution time and data transfer thanks to the PTC approach. As future works, in the current implementation, all expanded queries are executed sequentially. Executing them in parallel can significantly reduce execution times of SPARQL property path queries.

**Acknowledgments** This work is supported by the national ANR DeKaloG (Decentralized Knowledge Graphs) project, ANR-19-CE23-0014, CE23 - Intelligence artificielle.

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
