# OpenReview forum: "SaGe-Path: Pay-as-you-go SPARQL Property Path Queries Processing using Web Preemption"
_eswc-conferences.org/ESWC/2021/Conference/Poster_and_Demo_Track — ESWC2021 P&D_

### Official Review · AnonReviewer1 · 2021-04-14
**Ok demo of useful and solid work**

**Rating:** 7
**Confidence:** 5

**Review:**

The paper introduces the demo of SaGe-Path, an approach to optimize path query answering using server-side computation.

PROS

The technical work behind the demo is solid and relevant. It has been recently demonstrated that the approach outperforms alternatives with a clear impact on the support for queries not answered by public portals today. Optimizing path queries is also relevant for the semantic web community because it may bring some points to RDF over competing technologies optimized for graph traversal.

The paper is well written and well organized; it explains what the users will learn at the demo, the technical background, and its advantages.

CONS

The main disadvantage of the proposed demo is that it does not promise to be much exciting for the audience. Overall, optimal parameters’ setting is an important but technical issue that could be discussed in a paper.

SUMMARY

Although this does not look like a particularly exciting demo, showcasing basically parameters’ tuning and trade-offs that could be discussed in a paper, the work is solid and worth being advertised. It may be interesting to colleagues working on similar topics, who could better understand the proposed approach or to triple store vendors interested in the proposed solution.

Additional comments:

“For instance, SaGe “ → introduce SaGe that pops up out of the blue here
“it may require several quanta to terminate. “ → Quantum has not been defined explained before. I suggest clarifying this concept to make the paper self-contained.


**Anonymity:**

Yes, I would like my review to remain anonymous.

---

### Official Review · AnonReviewer4 · 2021-04-15
**The article presents an approach for processing online SPARQL property path queries called SAGE-Path. The work includes a visual interface which allows to understand the features of the method. Both, the method and the visual interface are very interesting, so they will be attractive for the audience.**

**Rating:** 9
**Confidence:** 4

**Review:**

Quality: High

Clarity: High

Originality: High

Significance: Very High

Pros
- The paper is well-written and structured.
- The examples used to describe the method (SAGE-Path) are very clear and easy to follow. Moreover, the description includes several comments that allow to understand the key features of the method.
- There exists a visual interface that allows to understand the features of SAGE-Path by showing information and statistics about the execution of a property path query.
- The article is accompanied by a video presenting the user interface.

Cons
- The explanation of the visual interface (shown in Figure 4) could be improved.
- It could be interesting to see experimental results with different types of queries.

**Anonymity:**

Yes, I would like my review to remain anonymous.

---

### Official Review · AnonReviewer3 · 2021-04-16
**The demo paper is a nice companion to the already accepted ESWC'21 conference paper. I suggest to accept it since the topic is relevant to the community, the presented solution is significant and  the demo scenario is clear.**

**Rating:** 9
**Confidence:** 4

**Review:**

The demo paper "SaGe-Path: Pay-as-you-go SPARQL Property Path Queries Processing using Web Preemption" addresses a topic that is of practical interest to the the Semantic Web community. The presented solution is significant and has been accepted to EWSC'21. The described demo scenario is clear. The narration and the language of the paper are also clear. So I suggest to accept the demo, it will be a good companion to the already accepted ESWC'21 paper and of interest to a wide audience.

**Anonymity:**

Yes, I would like my review to remain anonymous.

---

### Decision · Program_Chairs · 2021-04-19

Accept